

# Phase separation in DNA repair: orchestrating the cellular response to genomic stability

Juxin Deng[1,*], Zhaoyang Du[1,*], Lei Li[2], Min Zhu[3] and Hongchang Zhao[2]

[1] Department of Emergency Surgery, The First Affiliated Hospital of Bengbu Medical University, Bengbu, Anhui, China

[2] Department of Emergency Surgery, The First Affiliated Hospital of Bengbu Medical University, Institute of Emergency and Critical Care Medicine, Bengbu, Anhui, China

[3] School of Life Science, Anhui Agriculture University, Hefei, Anhui, China

[*] These authors contributed equally to this work.

## ABSTRACT

DNA repair is a hierarchically organized, spatially and temporally regulated process involving numerous repair factors that respond to various types of damage. Despite decades of research, the mechanisms by which these factors are recruited to and depart from repair sites have been a subject of intrigue. Recent advancements in the field have increasingly highlighted the role of phase separation as a critical facilitator of the efficiency of DNA repair. This review emphasizes how phase separation enhances the concentration and coordination of repair factors at damage sites, optimizing repair efficiency. Understanding how dysregulation of phase separation can impair DNA repair and alter nuclear organization, potentially leading to diseases such as cancer and neurodegenerative disorders, is crucial. This manuscript provides a comprehensive understanding of the pivotal role of phase separation in DNA repair, sheds light on the current research, and suggests potential future directions for research and therapeutic interventions.

## INTRODUCTION

The integrity of the genome is under constant threat from both endogenous and exogenous sources of DNA damage (*Harper & Elledge, 2007*; *He et al., 2024*). To counteract this, cells have evolved a sophisticated network of repair mechanisms that ensure the maintenance of genomic stability, a prerequisite for cell survival and the prevention of diseases such as cancer. Among these mechanisms, the DNA damage response (DDR) is a critical signaling pathway that detects lesions, signals their presence, and orchestrates their repair (*Harper & Elledge, 2007*; *Huang & Zhou, 2021*). Despite decades of research on the role of the DDR in genome stability maintenance, how numerous repair factors coordinate and function systematically remains unclear.

Recent insights suggest that the spatial and temporal organization of repair factors is not solely dependent on static protein-protein interactions but is dynamically regulated by phase separation mechanisms. A growing body of evidence suggests that liquid-liquid phase

Corresponding authors
Min Zhu, minzhu@ahau.edu.cn
Hongchang Zhao,
hozhao@bbmu.edu.cn

separation (LLPS), a fundamental principle governing the organization of biomolecules within cells, plays a crucial role in facilitating DNA repair.

LLPS facilitating the formation of dynamic, membraneless organelles, such as nucleoli, stress granules, and promyelocytic leukemia (PML) bodies (*Diaz-Moreno & De la Rosa, 2021*; *Jiang et al., 2020*; *Liu et al., 2024*; *Spruijt, 2023*). These organelles serve as hubs for biochemical activities, concentrating specific proteins and nucleic acids to increase reaction rates or sequester components away from the bulk cellular environment (*Maccaroni et al., 2022*; *Ng, Sielaff & Zhao, 2022*). In the context of the DDR, LLPS facilitates the formation of membraneless repair foci at DNA damage sites, concentrating repair proteins and nucleic acids, thereby enhancing repair efficiency (*Levone, Lombardi & Barabino, 2022*; *Mine-Hattab et al., 2021*; *Mine-Hattab, Liu & Taddei, 2022*). This emerging concept provides a new perspective on how DNA repair is orchestrated at the molecular level.

In this manuscript, we explore the mechanisms through which phase separation facilitates the concentration and coordination of repair factors, the regulatory aspects governing this process, and the consequences of its dysregulation. We aim to provide a comprehensive synthesis of recent advances, addressing key questions in the field. This review is intended for researchers in molecular biology, biophysics, and related fields who seek a deeper understanding of the molecular mechanisms governing LLPS in the DDR. By synthesizing recent advances and identifying open questions, we highlight how phase separation orchestrates genomic integrity and has attracted growing interest as a possible contributor to the identification of disease markers or therapeutic approaches, although its clinical utility remains to be fully validated.

## SURVEY METHODOLOGY DATABASES AND KEYWORDS USED

To summarize the role of LLPS in DNA genome stability, we conducted a systematic search using multiple scholarly databases, including PubMed and Google Scholar. The following keywords were used: "DNA repair" and "phase separation", "liquid-liquid phase separation", "genome stability", "biomolecular condensates", "DNA damage response", "post-translational modifications", and "DNA repair factors".

## CONCEPTUAL FRAMEWORK OF PHASE SEPARATION IN CELLULAR ORGANIZATION

At its core, LLPS refers to the process whereby a homogenous solution separates into two distinct liquid phases, each enriched with different components. This phenomenon is not merely a peculiarity of physical chemistry but also a fundamental principle that underpins the spatial organization within cells (*Diaz-Moreno & De la Rosa, 2021*; *Zhang et al., 2020*). In biological systems, LLPS facilitates the formation of membraneless organelles or biomolecular condensates, which provide a unique environment for specific biochemical processes to occur with increased efficiency and specificity. These condensates include stress granules and P-bodies, which play key roles in RNA processing, as well as the nucleolus, the central hub of ribosome biogenesis (*Jiang et al., 2020*; *Maccaroni et al., 2022*).

The driving forces behind LLPS in cells are multifaceted and involve a complex interplay of weak multivalent interactions among proteins and nucleic acids (*Wei et al., 2022*). These interactions include electrostatic and hydrophobic interactions, hydrogen bonding, and π–π interactions (*Ren et al., 2022*; *Vendruscolo & Fuxreiter, 2023*). The intrinsic disorder of many phase-separating proteins allows flexible engagement in multiple transient interactions, making them prime candidates for LLPS (*Ng, Sielaff & Zhao, 2022*). Moreover, the role of RNA not only as a genetic messenger but also as an active participant in phase separation has been increasingly recognized. RNA molecules or R-loops can act as scaffolds or clients within condensates, contributing to the specificity and dynamics of phase-separated entities (*He et al., 2023*; *Nozawa et al., 2020*; *Xu et al., 2024*).

The compartmentalization achieved through LLPS has profound implications for cellular organization and function. By concentrating specific molecules, these condensates can increase reaction rates, protect sensitive molecules from degradation, or sequester potentially harmful entities away from the interior of the cell. This compartmentalization is dynamic and reversible, allowing cells to respond rapidly to environmental cues or stress conditions (*Saito & Kimura, 2021*). Furthermore, the regulation of LLPS is tightly controlled by various post-translational modifications (PTMs), such as phosphorylation, methylation, and especially SUMOylation, which modulate the interaction affinities between phase-separating components and thus the properties of the condensates themselves (*Cheng, 2023*; *Li et al., 2022*; *Liu et al., 2022*; *Long et al., 2023*; *Wei et al., 2023*).

While LLPS is a critical regulatory mechanism in cellular physiology, its dysregulation is implicated in various diseases. The aberrant formation, persistence, or composition of biomolecular condensates can lead to pathological conditions, including neurodegenerative diseases and cancer (*Zhang et al., 2023*). In the context of cancer, alterations in the mechanisms governing LLPS can affect the behavior of oncogenes and tumor suppressors within phase-separated nuclear bodies, thereby influencing gene expression, DNA repair, and genome stability (*Tong et al., 2022*). Understanding the specific conditions under which LLPS contributes to disease is an active area of research with significant implications for the development of novel therapeutic strategies.

Despite the significant strides in understanding LLPS in terms of cellular organization, numerous questions remain. The precise molecular mechanisms governing the formation, regulation, and function of phase-separated condensates are still being elucidated. Advanced LLPS detection strategies, such as imaging techniques combined with biophysical and biochemical approaches, are uncovering the roles of LLPS in unprecedented detail. As research progresses, the challenge lies in translating this knowledge into therapeutic interventions that can correct the dysregulation of LLPS in diseases (*Tosolini et al., 2020*).

## MASTERING GENOMIC INTEGRITY: THE ROLE OF PHASE SEPARATION IN DNA REPAIR MECHANISMS

Our genomic DNA faces relentless assault from both internal and external sources. Endogenously, reactive oxygen species from cellular metabolism and external threats, such as ultraviolet radiation and chemical carcinogens, inflict various forms of DNA damage

(*Huang & Zhou, 2021*; *Thada & Greenberg, 2022*; *Zhu & Zheng, 2020*). This damage, ranging from single- and double-strand breaks to base modifications and crosslinks, challenges genomic integrity, necessitating precise repair mechanisms. To counteract these diverse types of damage, cells have evolved sophisticated repair systems to preserve genomic stability.

To achieve this, cells employ multiple specialized repair pathways, each tailored to address specific types of DNA damage. These include base excision repair (BER), nucleotide excision repair (NER), homologous recombination (HR), and non-homologous end joining (NHEJ) to maintain genomic integrity (*Lecca & Ihekwaba-Ndibe, 2022*; *Li & Xu, 2016*; *Li et al., 2020*).

The DNA repair process is not merely a simple collection of independent pathways but is hierarchically organized and dynamically regulated. For example, the MRN complex recognizes the DNA damage site, initiating the activation of ATM/ATR to amplify the damage signaling cascade, followed by the spread of γH2AX around the damage site and the recruitment of the mediator protein MDC1. Similarly, various repair pathways exhibit distinct hierarchy processes to execute the steps of recognition, recruitment, amplification, and repair. These sequential regulations are essential for coordinating the repair pathways in space and time, ensuring that the correct repair mechanism is employed for each type of damage. The choice between different repair pathways can be influenced by factors such as the phase of the cell cycle, the nature and extent of DNA damage, and the cellular context (*Jachimowicz, Goergens & Reinhardt, 2019*; *Li & Xu, 2016*; *Scully et al., 2019*). Furthermore, DNA repair is a highly dynamic process. Repair factors are not statically positioned but are dynamically recruited to and released from sites of damage, which depend on protein-protein interactions, post-translational modifications, the microenvironment, and phase separation (*Kong, Beckwitt & Van Houten, 2020*).

Among these regulatory factors, phase separation has recently emerged as a critical mechanism governing the spatiotemporal organization of DNA repair. Accumulating evidence suggests that phase separation facilitates the formation of biomolecular condensates, which concentrate repair factors and enhance repair efficiency. To date, there are four potential mechanisms through which phase separation enhances DNA repair efficiency:

## LLPS creates localized high concentrations of repair proteins

One of the primary ways LLPS enhances DNA repair is by facilitating the recruitment and retention of repair proteins at DNA damage sites, ensuring that repair machinery is readily available. For instance, FUS-dependent phase separation plays a crucial role in the early DDR by concentrating key repair factors such as KU80, NBS1, 53BP1, and SFPQ. In FUS-knockout cells, these proteins fail to accumulate at DNA breaks, leading to inefficient repair. Moreover, LLPS-inhibiting mutations and chemical disruption (1,6-hexanediol) prevent DDR foci formation, demonstrating that LLPS is essential for maintaining repair factor concentrations at DNA lesions (*Levone et al., 2021*).

## LLPS enhances the formation of repair complexes

Beyond simply localizing repair factors, LLPS also plays a crucial role in assembling multi-protein repair complexes. For example, RAP80 undergoes LLPS at DNA double-strand breaks (DSBs), driven by its intrinsically disordered region (IDR1) and further enhanced by Lys63-linked poly-ubiquitin chains, promoting multivalent interactions and condensate formation, which are essential for the recruitment of BRCA1-A complex. Mutation studies have shown that disrupting RAP80 LLPS, either by deleting its ubiquitin-interacting motifs (UIMs) or through 1,6-hexanediol treatment, significantly reduces BRCA1 recruitment, underscoring the necessity of LLPS for proper complex formation (*Qin et al., 2023*). Beyond RAP80, 53BP1 also utilizes LLPS to orchestrate multi-protein interactions at DNA damage sites. 53BP1 condensates act as scaffolds for downstream repair factors such as RIF1, REV7, and Shieldin, and disrupting this phase-separation process abolishes their recruitment, reinforcing LLPS as a universal mechanism that enhances repair efficiency and ensures spatiotemporal coordination of repair pathways (*Kilic et al., 2019*).

## LLPS prevents non-specific repair factor interactions, ensuring targeted DNA repair

In addition to recruiting repair factors, LLPS also play a critical role in maintaining repair specificity by restricting inappropriate repair factor interactions. For example, RNF168 phase-separates into nuclear condensates, sequestering 53BP1 and restricting its recruitment to DNA damage sites, thereby modulating non-homologous end joining (NHEJ) efficiency. Additionally, SENP1 functions as a deSUMOylase, reducing RNF168 LLPS to enable the proper recruitment of repair proteins, highlighting how LLPS contributes to repair pathway specificity and prevents aberrant factor interactions (*Wei et al., 2023*).

## The dynamic nature of LLPS regulates repair condensate assembly and disassembly

A key advantage of LLPS in DNA repair is its dynamic nature, allowing repair condensate to assemble and disassemble as needed, ensuring an efficient and controlled repair process. PARP1-generated poly(ADP-ribose) (PAR) chains initiate condensate formation, while PARG-mediated hydrolysis of PAR triggers condensate dissolution, providing a regulatory mechanism for repair site resolution (*Duan et al., 2019*). Additionally, SUMOylation and phosphorylation fine-tune the stability of repair condensates, ensuring that repair proteins are recruited only when necessary, thus preventing prolonged or excessive repair signaling (*Arnould et al., 2023*; *Chen et al., 2023*; *Mine-Hattab, Liu & Taddei, 2022*; *Spegg & Altmeyer, 2021*; *Spruijt, 2023*; *Stanic & Mekhail, 2022*; *Wang et al., 2023*).

These mechanisms illustrate how LLPS plays a multifaceted role in coordinating DNA repair, ensuring that repair factors are properly localized, assembled, and dynamically regulated to maintain genomic stability. Figure 1 shows how liquid–liquid phase separation (LLPS) dynamically orchestrates the formation and dissolution of DNA repair condensates. By concentrating repair proteins within localized compartments, LLPS promotes the assembly of multi-protein complexes, minimizes non-specific interactions, and enhances both the efficiency and accuracy of DNA repair. The circular structure in the figure

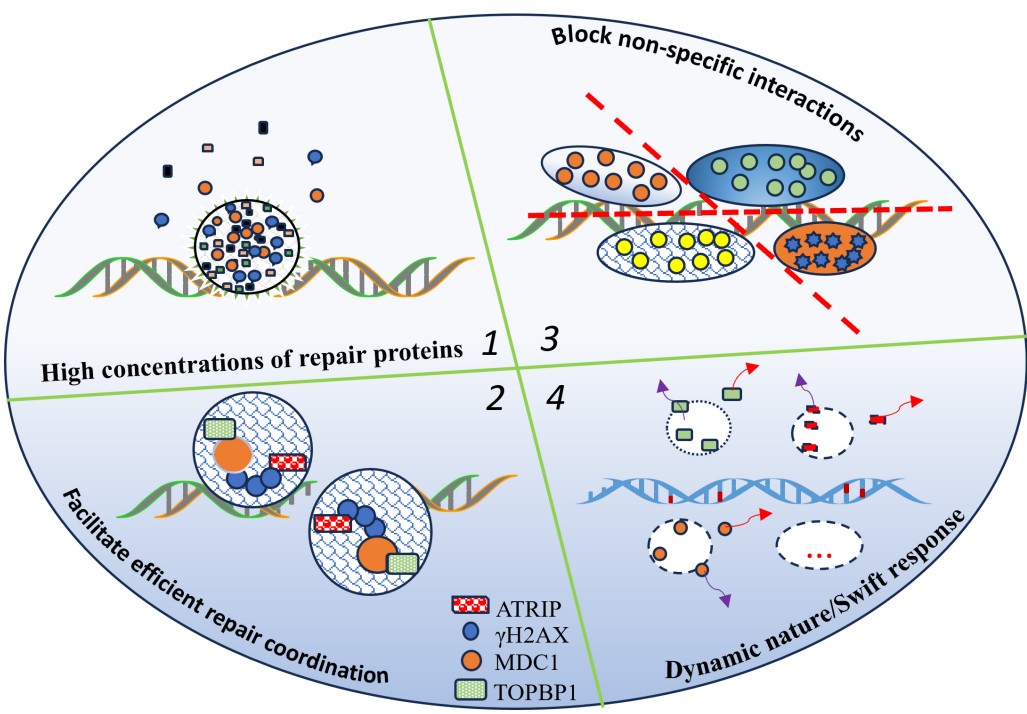

**Figure 1** **LLPS enables the dynamic formation and dissolution of repair condensates, optimizing DNA repair efficiency.** LLPS organizes and regulates the DNA repair machinery, thereby increasing the efficiency and accuracy of the repair process for the following reasons: (1) localized high concentrations of repair proteins; (2) facilitation of multi-protein complex assembly; (3) reduction of non-specific interactions; (4) LLPS enables the dynamic formation and dissolution of repair condensates, optimizing DNA repair efficiency; and the circular structure represents the LLPS condensate.

represents the phase-separated condensate where these molecular processes are spatially organized.

## PHASE SEPARATION IN DNA REPAIR: ORCHESTRATING GENOMIC INTEGRITY THROUGH DYNAMIC CELLULAR MECHANISMS

The process of DNA repair is a critical mechanism by which cells maintain genomic integrity and prevent diseases such as cancer (*Huang & Zhou, 2021*; *Li et al., 2020*; *Pessina et al., 2021*). The dynamic process of LLPS during the repair process involves the formation of phase-separated condensates and specialized microenvironments that serve as efficient "repair factories". The key to the formation of these condensates is multivalent interactions among repair proteins, RNA, and DNA. Proteins featuring intrinsically disordered regions (IDRs) or multiple interaction domains are particularly adept at undergoing phase separation, along with sophisticated post-translational modifications that form reversible complexes crucial for initiating and regulating repair. Additionally, physicochemical factors such as pH and salt concentration can modulate phase separation behavior by influencing intermolecular interactions, though their precise role in DNA repair condensates requires

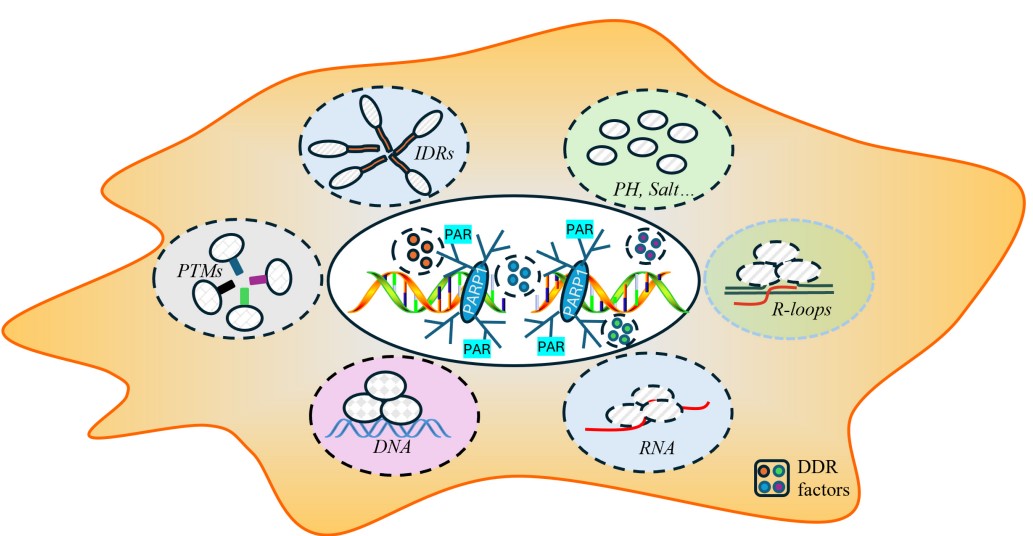

**Figure 2** Key factors influencing the formation of DNA repair conditions include intrinsically disordered regions (IDRs), post-translational modifications (PTMs), RNA-DNA hybrid structures, the local pH and ionic strength, and DNA and RNA molecules.

further exploration. Moreover, the damaged DNA itself, along with RNA molecules and specific scaffolding proteins, provides a structural framework that enhances the localization and stability of these complexes, ensuring that the repair machinery is precise where it is needed (*Alghoul, Basbous & Constantinou, 2023*; *Arnould et al., 2023*; *Diaz-Moreno & De la Rosa, 2021*; *Harami et al., 2020*; *Kilic et al., 2019*; *Nozawa et al., 2020*; *Singatulina et al., 2019*) (Fig. 2).

Importantly, thess phase-separated condensates are not static assemblies; rather, they exhibit liquid-like properties and viscoelasticity, allowing for fluid and responsive adaptations to the nature and extent of DNA damage (*Gao et al., 2022*). This adaptability is further regulated by post-translational modifications (PTMs) of repair proteins, which can influence both the assembly of repair-focused condensates and the activity of their constituent proteins, such as phosphorylation, acetylation, poly (ADP-ribosylation) (PAR), ubiquitination, methylation, and SUMOylation (*Cheng, 2023*; *Duan et al., 2019*; *Li et al., 2022*; *Liu et al., 2022*). These PTMs act as molecular switches that finely tune the repair process, dynamically adjusting the condensate composition in response to cellular signals and repair needs (*Cheng, 2023*; *Duan et al., 2019*; *Frattini et al., 2021*; *Long et al., 2023*; *Rhine et al., 2023*; *Wei et al., 2023*).

Despite significant advances in understanding the role of PTMs in regulating phase separation during DNA repair, several challenges remain. One major challenge is the complexity of PTM networks, where a single protein can undergo multiple modifications, each potentially having distinct effects on phase separation and DNA repair. Additionally, the transient and dynamic nature of biomolecular condensates poses technical challenges in the study of their formation and regulation in real time. Future research in this field is likely to focus on developing advanced imaging and biochemical techniques to study the

dynamics of PTMs and phase separation in live cells. Understanding the context-dependent effects of PTMs and revealing the crosstalk between different modifications will be crucial in comprehensively understanding how phase separation is regulated during the DNA repair process.

Beyond PTMs, the cellular microenvironment also plays a crucial role in regulating phase separation. Factors such as pH, ion concentration, and the presence of small molecules can significantly impact the formation and dissolution of condensates (*Alberti et al., 2018*; *Babinchak et al., 2020*; *Duan & Wang, 2024*; *Jin et al., 2022*; *Kathe, Novakovic & Allain, 2024*; *Li, Wang & Lai, 2023*; *Wiedner & Giudice, 2021*; *Yang et al., 2024*). These environmental cues serve as additional layers of regulation, ensuring that phase separation occurs under optimal conditions for efficient repair. This dynamic sensitivity allows cells to fine-tune repair processes in response to fluctuating intracellular conditions, enhancing both precision and efficiency in maintaining genomic integrity.

As DNA repair progresses, the dynamic nature of phase separation becomes evident. Shifts in PTM status can modulate protein interactions within the condensate, ultimately leading to its dissolution once repair is complete (*Li et al., 2022*; *Luo, Wu & Li, 2021*). This dissolution is essential for the timely termination of the repair process and prevents the potential for genomic instability due to lingering repair proteins at the DNA damage site. Feedback mechanisms that monitor repair status and adjust protein concentration and PTMs are key to this regulation, ensuring that phase separation occurs only when necessary.

Understanding the intricate dance of phase separation during DNA repair not only sheds light on fundamental cellular processes but also opens avenues for developing targeted therapies. By manipulating phase separation dynamics, it might be possible to enhance DNA repair where it is deficient or to inhibit it in diseases such as cancer, where repair processes are often hijacked for survival. The ongoing exploration of LLPS in DNA repair is not just a journey into the cell's inner workings; it is a step toward harnessing these mechanisms for therapeutic innovation, with the promise of novel treatments for a range of diseases rooted in genomic instability.

## IMPLICATIONS OF ABNORMAL PHASE SEPARATION IN DNA REPAIR

Dysregulated phase separation in the context of DNA repair can have significant implications for cellular function and overall organismal health (*Jiang et al., 2020*; *Li et al., 2024*; *Liu et al., 2021*; *Mathieu, Pappu & Taylor, 2020*; *Peng, Hsu & Wu, 2021*; *Zhang et al., 2022*). Understanding these implications is crucial, as they highlight the delicate balance that cells must maintain in their biochemical processes. The most direct implication of abnormal phase separation in DNA repair is genomic instability. When the formation or dissolution of phase-separated biomolecular condensates is not properly regulated, the efficiency and accuracy of DNA repair can be compromised, leading to the accumulation of DNA mutations, which are key factors in the development of cancer (*Basu, 2018*; *Sinkala, 2023*; *Wang, 2001*). For example, if a phase-separated condensate is aberrantly altered, it

might lead to inappropriate repair activities, increasing the risk of mutagenesis (*Jiang et al., 2020*).

While genomic instability is strongly associated with cancer, abnormal phase separation has also been linked to neurodegenerative diseases such as Alzheimer's disease, Parkinson's disease, and amyotrophic lateral sclerosis (ALS) (*Kanekura & Kuroda, 2022*; *Wang et al., 2021*). In these diseases, the abnormal phase separation of proteins, including those involved in DNA repair, can lead to toxic protein aggregates. These aggregates can disrupt cellular functions and lead to neuronal death.

Beyond its role in cancer and neuroprotection, DNA repair is also crucial for maintaining a functional immune system (*He et al., 2024*; *Lin, Tang & Zheng, 2022*; *Pan et al., 2024*; *Tong et al., 2024*), especially in the development and diversification of B and T cells (*Bassing, Swat & Alt, 2002*). Abnormal phase separation could affect the process of V(D)J recombination, which is essential for the generation of antibody diversity. Recent studies indicate that phase separation plays a key role in V(D)J recombination by organizing recombinase proteins like RAG1/RAG2 into dynamic condensates, which enhance DNA cleavage and repair efficiency. Disruptions in these phase-separated compartments could impair antigen receptor diversity, potentially leading to immunodeficiency or autoimmunity (*Lin, Tang & Zheng, 2022*; *Xiao, McAtee & Su, 2022*).

Similar to its role in immune function, phase separation in DNA repair is also crucial for neural development. One example involves FUS, a phase-separating RNA-binding protein implicated in DNA damage repair, whose mutations have been associated with neurodevelopmental abnormalities and amyotrophic lateral sclerosis (ALS) (*Moens et al., 2025*). Normally, FUS undergoes liquid-liquid phase separation (LLPS) to form biomolecular condensates at DNA damage sites, facilitating the recruitment of key repair factors (*Levone et al., 2021*). However, ALS-linked mutations alter the phase behavior of FUS, leading to its cytoplasmic mislocalization and the sequestration of other RNA-binding proteins, such as fragile X mental retardation protein (FMRP), into aberrant condensates (*Birsa et al., 2021*). This mislocalization disrupts normal RNA metabolism and impairs translation, particularly in motor neurons.

Another key player in phase separation and DNA repair is TDP-43. Like FUS, TDP-43 is essential for RNA metabolism and DNA repair but undergoes pathological phase separation in neurodegenerative conditions (*Mitra et al., 2019*; *Provasek et al., 2024*; *Song, 2024*). Normally, TDP-43 forms functional biomolecular condensates to facilitate transcriptional regulation and DNA damage response. However, in ALS/FTD, mutations disrupt its phase behavior, leading to mislocalization and aggregation in the cytoplasm, which not only disrupts RNA homeostasis but also impairs genome stability, particularly in proliferative neural progenitor cells, where unresolved DNA damage can lead to developmental abnormalities and predispose neurons to degeneration. Loss of nuclear TDP-43 results in widespread transcriptional defects, further compromising DNA integrity and accelerating cellular dysfunction. Studies have shown that TDP-43 pathology extends beyond ALS/FTD, with implications in Alzheimer's disease, Parkinson's disease, and Huntington's disease, highlighting a broader link between phase separation dysregulation, genome instability, and aging-related neurodegeneration.

Given the profound impact of abnormal phase separation on genomic stability and disease progression, targeting phase separation with small molecules has emerged as a promising therapeutic strategy. For instance, 1,6-hexanediol has been shown to modulate phase separation of DNA repair proteins, suggesting a potential avenue for correcting defective repair mechanisms in diseases like cancer and neurodegeneration (*Ming et al., 2019*).

While the potential for targeted therapies is promising, there are significant challenges to consider. One major challenge lies in the dynamic and complex nature of phase separation. As previously discussed, DNA repair condensates are highly regulated by intricate networks of protein-protein interactions and post-translation modifications. Targeting these processes requires exceptional specificity and precise timing, as disrupting normal phase separation could lead to unintended consequences, such as impairing essential repair functions.

Despite these challenges, abnormal phase separation in DNA repair is being explored as a possible contributor to disease pathogenesis. Certain misregulated proteins or PTMs associated with phase separation have shown correlations with cancers or neurodegenerative conditions, raising the possibility that, with further study, they may inform future diagnostic or prognostic strategies.

In summary, the implications of abnormal phase separation in DNA repair span various biological processes and diseases. The consequences of abnormal phase separation, ranging from cancer and neurodegenerative diseases to immune dysfunction, are far-reaching. Understanding these implications deepens our knowledge of cellular biology. It also guides the development of therapies and diagnostic tools, emphasizing the importance of this research.

## KEY QUESTIONS OF LLPS AND FUTURE DIRECTIONS

### Current understanding of LLPS in DNA repair

The exploration of phase separation in DNA repair has yielded significant insights, revealing that it is a fundamental mechanism that orchestrates the efficient and accurate repair of DNA. Key findings include the following: (1) Research has elucidated how phase separation facilitates the concentration and coordination of repair factors at DNA damage sites, enhancing repair efficiency. (2) Studies have identified various proteins and molecular interactions that drive the formation of phase-separated repair condensates. (3) Advances have been made in understanding how post-translational modifications and the cellular microenvironment influence phase separation in DNA repair (Table 1).

### Unresolved mechanistic questions

Despite these advances, several key questions and debates remain: (1) Mechanistic details: The precise molecular mechanisms by which phase separation contributes to different DNA repair pathways are not fully understood. (2) Regulation complexity: The complexity of how phase separation is regulated in different cellular contexts and responses to various types of DNA damage is still unknown. (3) Pathological consequences: The detailed role

**Table 1  LLPS of DNA repair factors and their function in DNA damage repair.**

| DNA repair factor | Contribute the LLPS | Function in DDR | Reference |
|---|---|---|---|
| MRNIP | IDR | Promote MRN complex binding to the DSBs, Activating ATM signaling, DSB sensing, End-resection, HR | *Wang et al. (2022)* |
| 53BP1 | DilncRNAs, oligomerization domain, BRCT domains | p53 stabilization, create localized environments for DNA repair, DNA damage induced cellular senescence, maintenance of heterochromatin integrity and genome stability | *Bleiler et al. (2023)*, *Ghodke et al. (2021)*, *Oda et al. (2023)* and *Zhang et al. (2022)* |
| RPA32 | ssDNA, N-IDR, Phosphorylation | BTR complex (BLM–TOP3A–RMI) and its associated proteins enrichment | *Spegg et al. (2023)* |
| RNF168 | SUMOylating | Restricts the recruitment of RNF168 to DNA damage sites, reduces H2A ubiquitination, restrains 53BP1 condensates, impairs NHEJ | *Wei et al. (2023)* |
| RAD52 | Petite DIMs | Drive DNA repair center assembly | *Oshidari et al. (2020)* |
| RAP80 | IDR, Ubiquitination | BRCA1 enrichment | *Qin et al. (2023)* |
| TOPBP1 | Phosphorylation, BRCT domain | Amplifies ATR activity, Replication control, Checkpoint, HR | *Frattini et al. (2021)* |
| miRISC | cirRNA | RAD51 recruitment | *Wang et al. (2023)* |
| NONO | RNA-binding domain, low-complexity domain (LCD) | DNA-Pk activation, NHEJ | *Fan et al. (2021)* |
| FUS | RNA-binding domain, low-complexity domain (LCD) | HR and NHEJ, γH2AX foci formation, proper assembly of DSB repair complexes | *Levone et al. (2021)*, *Reber et al. (2021)*, *Rhoads et al. (2018)* and *Sukhanova et al. (2022)* |
| DDX3X | PARylation | NHEJ, RNA binding proteins recruitment | *Cargill et al. (2021)* |

of dysregulated phase separation in disease pathogenesis, particularly in specific types of cancer and neurodegenerative disorders, requires further investigation.

## The role of LLPS in DNA damage foci

Additionally, one critical issue that must be addressed is the correlation between LLPS and its close association with the formation of DNA damage foci at the site of DNA damage. Undoubtedly, both LLPS and DNA damage foci are dynamic processes that primarily occur at the damage site. Upon DNA damage, these foci rapidly assemble to concentrate key repair factors through mechanisms such as protein-protein interactions, post-translational modifications, and potentially LLPS, thereby facilitating efficient repair. As the damage is resolved, these structures disassemble, ensuring that the repair process remains transient and tightly regulated.

Notably, several DNA-repair factors, such as RPA, 53BP1, RNF168 and RAD51, not only form DNA damage foci but also undergo LLPS, further concentrating repair proteins and signaling molecules. This phase separation enhances the efficiency and specificity of the repair process by compartmentalizing key components within a localized region. However, it is important to note that not all repair factors undergo LLPS. Some repair proteins are recruited to foci without undergoing phase separation, highlighting the complexity of the

repair process. The mechanisms regulating these processes, including how certain repair factors are selected for LLPS and the coordination between foci formation and disassembly, remain complex and are still not fully understood. Unraveling the regulatory network of LLPS in DNA damage repair is a critical area of ongoing research, as understanding these mechanisms could provide insights into the molecular basis of diseases related to genome instability, such as cancer and aging.

## Future directions

Future research in this field can take multiple directions: (1) advanced molecular and imaging techniques could provide deeper insights into the dynamics and regulation of phase-separated repair condensates, and (2) the development of large-scale screening strategies for LLPS factors *in vivo* is essential to aid researchers in studying specific biological processes. A recent high-throughput study systematically identified a diverse set of phase-separating proteins *in vivo*, revealing over 1,500 endogenous biomolecular condensates, including 538 previously unreported ones. By leveraging density gradient ultracentrifugation and quantitative mass spectrometry, the study provided a proteome-wide map of phase-separating proteins and demonstrated their dynamic regulation in response to cellular conditions. These findings mark a significant step forward in the study of biomolecular condensates, paving the way for future strategies to explore their physiological and pathological roles (*Li et al., 2024*). (3) Integrating findings from phase separation research with broader aspects of cell biology, such as metabolism and cellular signaling, could offer a more holistic understanding of DNA repair processes. (4) While still an early stage, efforts to translate LLPS-related findings into therapeutic approaches, such as modulateing phase separation processes in the context of cancer or neurodegenerative diseases, represent an emerging area of interet. Further mechanistic and preclinical studies are needed to evaluate the feasibility and specificity of such strategies.

The insights gained from studying phase separation in DNA repair hold significant potential for therapeutic interventions: (1) Developing drugs that modulate the formation or function of repair condensates could be a novel strategy in cancer therapy or in mitigating the effects of neurodegeneration. (2) Components involved in phase separation could serve as biomarkers for diseases characterized by DNA repair defects, aiding in early diagnosis and personalized treatment strategies.

## CONCLUSION

Phase separation has emerged not only as a biophysical phenomenon but also as a vital orchestrator in the cellular response to DNA damage, enhancing the efficiency and accuracy of repair mechanisms. Through the concentration and coordination of repair factors, phase-separated condensates ensure that cells can swiftly and effectively respond to and repair DNA damage, thus safeguarding against genomic instability and the potential onset of disease.

The exploration of the dynamics of phase separation has opened new vistas in understanding how cells regulate complex molecular interactions in response to DNA damage. The intricate interplay between molecular biochemistry and physical processes

in terms of cellular function has also been highlighted. The insights gained from studying phase separation in DNA repair have profound implications, especially in diseases such as cancer and neurodegenerative disorders, where DNA repair mechanisms are often compromised.

In the future, the intersection of phase separation and DNA repair will present fertile ground for research, offering potential avenues for innovative therapeutic strategies. Targeting phase separation processes to increase DNA repair efficiency or sensitize cancer cells to treatments holds considerable promise. Furthermore, the components and regulators of phase-separated repair condensates could serve as valuable biomarkers for disease diagnosis and prognosis.

In conclusion, the study of phase separation in DNA repair not only contributes to our fundamental understanding of cellular biology but also opens doors to novel approaches in disease treatment and management. As research continues to advance, it holds the promise of yielding transformative strategies in the fight against diseases marked by DNA repair dysfunction.

### Funding
The authors received no funding for this work.

### Competing Interests
The authors declare there are no competing interests.

### Author Contributions
- Juxin Deng conceived and designed the experiments, performed the experiments, analyzed the data, authored or reviewed drafts of the article, and approved the final draft.
- Zhaoyang Du performed the experiments, prepared figures and/or tables, authored or reviewed drafts of the article, and approved the final draft.
- Lei Li performed the experiments, analyzed the data, prepared figures and/or tables, and approved the final draft.
- Min Zhu conceived and designed the experiments, analyzed the data, prepared figures and/or tables, authored or reviewed drafts of the article, and approved the final draft.
- Hongchang Zhao conceived and designed the experiments, prepared figures and/or tables, authored or reviewed drafts of the article, and approved the final draft.

### Data Availability
This is a literature review.

### Supplemental Information
Supplemental information for this article can be found online at http://dx.doi.org/10.7717/peerj.19402#supplemental-information.

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
