# Peer review of "Phase separation in DNA repair: orchestrating the cellular response to genomic stability"

_PeerJ, doi:10.7717/peerj.19402_

## Round 0.1 · original submission · Major Revisions

Please address the concerns of all reviewers and amend the manuscript accordingly.

·

Basic reporting

Pls see the detailed comments.

Experimental design

Pls see the detailed comments.

Validity of the findings

Pls see the detailed comments.

Additional comments

This review presents a timely and relevant discussion on the role of liquid-liquid phase separation (LLPS) in DNA repair, emphasizing its regulatory impact on genome stability and disease progression. Several issues must be addressed, including grammatical errors, clarity of figures and captions, and some redundant or unclear phrasing.
The study is interesting. However, some issues need to be addressed before publication; the language and grammar errors need to be edited, and some important references are missing, such as PMID: 39192113; PMID: 39088918; PMID: 38655783; PMID: 38630271; PMID: 33749478; PMID: 32998978; PMID: 36016945.

Major concerns:

Clarity and Organization
1. The manuscript lacks clear subheadings in some sections (e.g., Survey Methodology, Key Questions of LLPS and Future Directions). Adding structured subheadings would improve readability and logical flow.
2. Several sections contain redundant explanations of LLPS and DNA repair pathways, leading to unnecessary repetition. Consider streamlining these discussions.
3. The manuscript has frequent grammatical and typographical errors that hinder readability. Below are examples of errors that must be corrected:
1, Sentence structure errors:
Line 28: "How the tons repair factors coordinate and work systematically is still a mystery to the researchers." Correction "How numerous repair factors coordinate and function systematically remains unclear".
2, Repetitive phrasing:
Line 34: "LLPS is a fundamental principle governing the organization of biomolecules within the cell, facilitating the formation of dynamic, membrane-less organelles..." Correction: Remove redundancy: "LLPS governs biomolecule organization, facilitating dynamic, membrane-less organelle formation."
3, Inconsistent verb tense:
Line 42: "Up to now, research has increasingly focused on unraveling the mechanisms underlying phase separation..."
Correction: "Recent research has focused on unraveling the mechanisms of phase separation..."
4, Comma splices and run-on sentences:
Line 191 "Understanding these implications not only deepens our knowledge of cellular biology but also guides the development of potential therapies and diagnostic tools, highlighting the importance of this area of research in molecular and cellular biology."
Correction: "Understanding these implications deepens our knowledge of cellular biology. It also guides the development of therapies and diagnostic tools, emphasizing the importance of this research. "
5, Grammatical Incorrect
Line 115 Missing verb: The phrase "but a hierarchically organized and dynamically regulated" lacks a verb, making the sentence structurally incomplete.
Correction: The verb "is" should be added for proper parallelism: "The DNA repair process is not merely a simple collection of independent pathways but is hierarchically organized and dynamically regulated."
Line 120 Since the other words ("recruitment," "amplification," "repair") are in noun form, "recognize" should be changed to "recognition" for consistency.
Line 126 The original phrase "which depend on" made it sound like "sites of damage" are the subject, but you are referring to "Repair factors." Changing it to "depending on" makes it grammatically correct.
6, Typographical errors:
Line 137: "Overall, the role of phase separation in DNA repair is a testament..."
Correction: "Overall, phase separation plays a critical role in DNA repair."
4, Figures and captions
The figures are not well labeled and explained. Specific issues:
Figure 1 Caption: "LLPS allows for the rapid assembly and disassembly of repair complexes."
Correction: "LLPS enables the dynamic formation and dissolution of repair condensates, optimizing DNA repair efficiency."
2, Figure 2 Caption: "Key factors Contributing to/Influence on the formation..." Correction: "Key Factors Influencing the Formation of DNA Repair Condensates."

Minor concerns:
1. Table Formatting Issues
The uniform font size and column alignment in Table 1 were ensured for better readability.
2. Overuse of Long Sentences
Some sentences exceed 40+ words—break them into shorter, digestible sentences for clarity.

Reviewer 2 ·

Basic reporting

The writing of the article was done well. It was written in clear professional English. Good amount of background was provided. This review covers the novel research area, which could be of high potential for new discovery in several related fields. The selected topic is unique, although not very new. There are several published articles on the LLPS and its cellular impacts out there.

Experimental design

This review paper research primary information from other publication.

Validity of the findings

Not applicable.

Additional comments

The story is potentially interesting. However, at it current form, it is lacking the scientific detail of the topic. The story tried to convince readers on the importance of LLPS on DNA repair, without describing the mechanism. Infact, there is no information offered from the authors.
To make this point clear; the authors coined in 4 possible roles of LLPS in DNA repair:
"1, By creating localized high concentrations of repair proteins, phase separation ensures that all necessary components are readily available at the damage site; 2,facilitates the assembly of multi-protein complexes, allowing for efficient coordination of the sequential steps in the repair process; 3, The localized environment reduces the likelihood of repair factors engaging in non-specific interactions, thus preserving their functional integrity; 4, The dynamic nature of phase-separated condensates allows for rapid assembly and disassembly, enabling a swift response to DNA damage and subsequent resolution once repair is complete. "
There is no detail on these 4 possible roles. This seems like expert comments. However, I believe that it is required to support the written comments with cited information.

The article suffers from the same pattern of writing in several other places.

Another area as example:
Line 203-213: It is only speculation. The authors need to specify and explain the writing there.

Please consider revise all of these points and beyond.

·

Basic reporting

In this review manuscript, Deng and collaborators describe the roles of liquid-liquid phase separation in DNA damage repair. Although some important and recent topics are covered, the manuscript lacks depth and many important aspects are lacking. Moreover, authors should describe specific findings from individual papers cited, as many concepts are still under discussion and authors present them as if they are already a common agreement among researchers in the field. There several points that need attention, listed below.

1- There is a problem with this manuscript's title. "Orchestrating Cellular Response to Genomic Instability". The authors probably meant "Genomic Stability.
2- There are asterisks in the two last names, but only one email for correspondence is displayed.
3- Lines 32-34 are repetitive.
4- Lines 39-49 "organelles, concentrating specific proteins and nucleic acids of repair foci" need rephrasing.
5- Lines 65-67 need rephrasing. Neither P-bodies nor stress granules have roles in RNA processing to nucleoli.
6- Line 118: "spread of rH2AX" - the authors meant to write gammaH2AX.
7- Line 139: Figure one is cited there, but with no link to the text. The reader should know what this figure will show, and a short description is needed within the text.
8- Same for Figure 2 in line 152.
9- Could the authors clarify where the findings described in lines 197-198 are published?
10- The authors try to shortly describe the roles of LLPS in immunity (lines 203-206). The description provided is, however,vague and does not really pinpoint the roles that LLPS play. Also, these sentences seem disconnected from the rest of the manuscript.
11- Lines 208-213: Authors start describing that DNA damage is more likely to happen in rapidly dividing cells, and in development, and this can correlate to early onset diseases associated with aging. Can the author provide more details and example?
12- Lines 217-221 are very vague and far from reality. LLPS being used as biomarkers for diagnostic or prognostic tools: how can this be obtained? There is no literature reporting this and the authors should stick to what is published, or give a better rationale.
13- Line 236: Table 1 should be better described and integrated in the text.
14- Line 266: authors give hints about a publication without properly describing their findings.
15- The manuscript would benefit from language revision, as for example in lines 64 ("which provide"), 116 ("the MRN complex recognize"), 137 ("overall" - should be capital), etc.
16- Figure 1: describe all the steps (1-4) individually in the figure legends.
17- Figure 2: description should be rephrased. Also, authors describe "R-loops and RNA-DNA hybrid structures", which are actually the same thing. Salt, pH are shown in figure, but not described anywhere else in the manuscript.

Experimental design

No comment.

Validity of the findings

Although the authors mention some interesting topics, this review lacks depth, which might compromise its impact. Moreover, proposing LLPS as a potential biomarker is very vague, and authors do not propose how this can be obtained.

---

## Round 0.2 · Minor Revisions

Please address the remaining concerns of reviewer #3 and amend the manuscript accordingly.

·

Basic reporting

This is a revision; they have addressed all my concerns, and now it is acceptable.

Experimental design

This is a revision; they have addressed all my concerns, and now it is acceptable.

Validity of the findings

This is a revision; they have addressed all my concerns, and now it is acceptable.

Additional comments

This is a revision; they have addressed all my concerns, and now it is acceptable.

Reviewer 2 ·

Basic reporting

The authors have addressed all of my comments. I do not have other comments and concerns.

Experimental design

No other comments.

Validity of the findings

No other comments.

Additional comments

No other comments.

·

Basic reporting

Deng and collaborators present a revised and improved version of the original manuscript. The structure of manuscript is now easier to follow, and it is more informative. The authors have addressed most of my previous concerns.
The authors, however, did not appropriately tune down their statement that LLPS represents a potential biomarker or therapeutic target (e.g. line 55, 305, 367) for certain disorders. For that reason, my previous concern that certain statements described by the authors are misleading and far from reality remains. For example, the authors proposed that alterations of LLPS-related PTM could be used in diagnostics, but these are generally not specific for LLPS. The study of the biophysical properties of mutated proteins and whether this impairs their LLPS capacity seems to be a more feasible approach, but it isn't cited.
Moreover, despite the proposal that the development of drugs that modulate the function of repair condensates is a tempting idea, it is not that easy to achieve, or to confirm whether it is really acting through LLPS modulation, but not other mechanisms.

Overall, I believe that the manuscript would benefit from either: 1) tuning down the statements to prevent misleading the readers; or 2) giving precise examples of how LLPS could be used for diagnostic or drug development (e.g., which PTM would be assessed, or how to prove that the drugs' improvements are through LLPS improvement, etc).

Some other small changes are required:
- Correct line 39: "LLPS facilitating".
- Correct cases where references are displayed after a period (e.g., line 45).
- Instead of just putting a title for the figures on the text (e.g., line 182 for Fig. 1), describe them better (e.g., Figure 1 below describes how the dynamic formation...).

Experimental design

no comment

Validity of the findings

no comment

Additional comments

no comment

---

## Round 0.3 · accepted · Accept

All remaining concerns of the reviewer were addressed, and the revised manuscript is acceptable now.